# Biological Control of *Fusarium oxysporum* f. sp. *cubense* Tropical Race 4 Using Natively Isolated *Bacillus* spp. YN0904 and YN1419

**DOI:** 10.3390/jof7100795

**Published:** 2021-09-24

**Authors:** Huacai Fan, Shu Li, Li Zeng, Ping He, Shengtao Xu, Tingting Bai, Yuling Huang, Zhixiang Guo, Si-Jun Zheng

**Affiliations:** 1Yunnan Key Laboratory of Green Prevention and Control of Agricultural Transboundary Pests, Agricultural Environment and Resources Institute, Yunnan Academy of Agricultural Sciences, 2238 Beijing Road, Kunming 650205, China; hcfan325@126.com (H.F.); lishukm@yeah.net (S.L.); heping_superv@163.com (P.H.); sj_xushengtao@163.com (S.X.); baiting8797@126.com (T.B.); yy189@21cn.com (Y.H.); zhixiangg@163.com (Z.G.); 2Bioversity International, 2238 Beijing Road, Kunming 650205, China

**Keywords:** *Foc* TR4, *Bacillus amyloliquefaciens*, *B. subtillis*, TR4-inhibitory, biocontrol, biocontrol mechanism

## Abstract

Fusarium wilt of banana (FWB) is the main threatening factor for banana production worldwide. To explore bacterial biocontrol resources for FWB, the antagonistic effective strains were isolated from banana-producing areas in Yunnan Province, China. Two isolates (YN0904 and YN1419) displaying strong antagonism against Tropical Race 4 (TR4) were identified from a total of 813 strains of endophytic bacteria. TR4 inhibition rates of YN0904 and YN1419 were 79.6% and 81.3%, respectively. By looking at morphological, molecular, physiological and biochemical characteristics, YN0904 was identified as *Bacillus amyloliquefaciens*, while YN1419 was identified as *B. subtillis*. The control effects of YN0904 and YN1419 on TR4 in greenhouse experiments were 82.6% and 85.6%, respectively. Furthermore, YN0904 obviously promoted the growth of banana plantlets. In addition, biocontrol marker genes related to the biosynthesis of antibiotics synthesized and auxin key synthetase genes could be detected in YN0904. Surprisingly, the marker gene *sboA* could be exclusively detected in YN1419, while other marker genes were all absent. Molecular characterization results could provide a theoretical basis for expounding the biocontrol mechanisms of these two strains. We concluded that natively antagonistic strains derived from local banana plantations could provide new biological control resources for FWB.

## 1. Introduction

Banana (*Musa* spp.) is a globally important (sub) tropical economic and food crop, grown in more than 130 countries [1,2]. China has large banana-production areas, mainly in Guangxi, Hainan, Guangdong, Fujian, Taiwan and Yunnan provinces [3]. Yunnan Province is one of the original centers of the banana and has a long history of banana cultivation. Banana production supports a key income for ethnic groups in remote areas and offers a route to alleviate poverty, playing an important economic role in regional rural society.

Fusarium wilt of banana (FWB) is a globally devastating disease caused by *Fusarium oxysporum* f. sp. *cubense*; Tropical Race 4 (TR4) is the most virulent race [4]. Being a typical soil-borne disease, TR4 is very difficult to control [5,6]. Resistant breeding is usually the most effective method; however, it is difficult to deploy conventional cross-breeding methods due to the main commercial cultivars are triploid and seedlessness [7,8]. At present, there are no fully TR4-resistant cultivars available. Some partially resistant cultivars underperform; these are more vulnerable to local conditions and possess unstable and inferior agronomic traits [1]. There are also several studies of fungicidal screening for TR4 control [9,10,11], but most of them only work in indoor or pot experiments and are rarely effective in controlling soil-borne diseases in fields. Furthermore, chemical agents are a main source of environmental pollution [12]. Some complementary control measures, such as rotation with Chinese leek, sugarcane and maize, have been reported to reduce FWB incidence [13,14], but the rotation takes a long period; hence, this is not a realistic option in banana perennial cropping systems. Using beneficial and antagonistic microorganisms for biological control offers a potential, environmentally friendly, and effective approach to control FWB. There are many studies on using effective biocontrol agents to control TR4, such as *Trichoderma harzianum* [15], *Streptomyces* and *Bacillus* [16,17,18,19]. Nel et al. [20] also found that non-pathogenic *Fusarium oxysporum* could potentially be used for FWB control.

Many biocontrol microorganisms can produce a variety of secondary metabolites in their growth processes, which have many functions, such as inhibiting the growth of pathogens, promoting plant growth, producing some lipopeptides such as surfactin and fengycin, and inducing systemic resistance of plants [21,22,23]. The biosynthetic pathways of antimicrobial metabolites by *Bacillus* include the ribosomal pathway, non-ribosomal pathway and polyketide pathway [21]. Lipopeptide antibiotics are synthesized by non-ribosomal pathways such as surfactin, fengycin, iturin and bacillibactin; polyketides include bacillibactin, bacillaene and difficidin. Subtilosin A synthesized by the ribosomal pathway has been proved to play an important role in the antagonistic process of *Bacillus*, and the key synthase genes for these metabolites are also considered to be biocontrol marker genes [21,24]. In addition, some *Bacillus* have the ability to promote plant growth, such as the synthesis of plant auxin indole-3-acetic acid (IAA) and its analogs [25].

In this study, we would like to explore the possibility of using natively isolated *Bacillus* strains which were collected from banana plants from different banana-production areas in Yunnan for FWB antagonism. A pot experiment and PCR characterization with 12 biocontrol marker genes and an auxin synthesis gene *ysnE* will be implemented for two selected *Bacillus* strains, which are helpful to initially assess the biocontrol potential of these antagonistic strains. This work aims to enrich biocontrol resources and lays a foundation for comprehensive control on FWB.

## 2. Materials and Methods

### 2.1. Materials

#### 2.1.1. The Culture Media Was Used as Follows

Medium PDA: Potato—200 g, Glucose—20 g, Agar—15 g, Water—1 L, pH = 7. PDB has the same gradient as PDA without agar.

Medium NA: Peptone—10 g, Beef Extract—3 g, Sodium Chloride—5 g, Agar—15 g, Water—1 L, pH = 7.

Medium LB: Tryptone—10 g, Yeast Extract—5 g, Sodium Chloride—10 g, Water—1 L, pH = 7.

#### 2.1.2. Pathogen

The tested pathogen is TR4 strain 15-1, which was isolated, identified and preserved by the Banana Research Team, Agricultural Environment and Resources Institute, Yunnan Academy of Agricultural Sciences. The TR4 strain (15-1) was isolated from banana cultivar Brazilian (Cavendish, AAA) in the planting field of Xishuangbanna in 2015. This strain is highly virulent and pathogenic to banana [26].

#### 2.1.3. Banana Cultivar

The banana cultivar Brazilian (Cavendish, AAA) was used in the greenhouse pot experiment, which was propagated by the aforementioned Banana Research Team.

### 2.2. Isolation and Screening of Antagonistic Endophytic Bacteria against FWB

#### 2.2.1. Isolation of Endophytic Bacteria

Since July 2009, healthy banana pseudostems and symptomless banana pseudostems (already naturally infected with both FWB and bacterial soft rot (*Erwinia chrysanthemi*)) were collected from different banana planting orchards in Yunnan Province. Isolation method of endophytic bacteria in banana pseudostems was as follows:

(i).Washing the collected pseudostems with sterile water, then sterilizing with 75% ethanol by spraying after drying naturally;(ii).The 2–3 cm middle tissue of pseudostems was extracted with a sterile blade, soaked in 75% ethanol for 30 s–1 min, then transferred in 0.1% mercuric chloride for 30 s–1 min, rinsed 2–3 times with sterile water, and ground into sap with a sterile pestle and mortar;(iii).Streaking the sap on the NA medium and culturing it at 37 °C for 48 h. Then, single colonies with different morphologies were selected for purification. The isolated strains were stored at −80 °C with 50% glycerin.

#### 2.2.2. Screening of Antagonistic Endophytic Bacteria against TR4

Primary screening was carried out after isolation. TR4 strain 15-1 was cultured at 28 °C for 7 days on PDA, and a 5 mm diameter slice was transferred to the center of a new PDA medium using a sterilized puncher, and then the isolated bacteria were dipped with an inoculation loop, which was 25 mm on the side of plate. After culturing at 28 °C for 7 days, the results were observed and recorded. Those strains with inhibition widths of more than 3 mm were selected for secondary screening.

The secondary screening of antagonistic bacteria was carried out according to the method of dual-culture test described by Li and He et al. [21]. The TR4 was inoculated in the center and antagonistic bacteria were inoculated at four points on cross-line graticule. The conditions of dual-culture test (including temperature and culture time) were consistent with that of initial primary screening. The PDA medium only inoculated with TR4 was used as a control, and each treatment was replicated three times. After being cultured at 28 °C for 7 days, inhibition effect results were assessed. The diameter of the TR4 colony was determined by the cross method. Inhibition rate (%) = (TR4 colony diameter of control group—TR4 colony diameter of treated group)/TR4 colony diameter of control group × 100.

### 2.3. Effect of Antagonistic Strains on Morphology of TR4 Hyphae

TR4 was cultured with antagonistic bacteria from secondary screening for 7 days at 28 °C using the dual-culture method. The fresh hyphae of TR4 close to the antagonistic bacteria as described by Li and He et al. [21] were selected using a sterilized toothpick and transferred to a temporary microscope slide. The effects of antagonistic bacteria on the growth of TR4 hyphae were observed using a scanning electron microscope (ZEISS Sigma 300, Berlin, Germany). The new TR4 hyphae of the single cultured were used as a control.

### 2.4. Identification of Antagonistic Strains

#### 2.4.1. Morphological Observation

The antagonistic strains were cultured on NA medium at 37 °C for 24 h, then the colony morphology, transparency, color, and other characteristics were observed and recorded. Morphologies of antagonistic strains were observed using a scanning electron microscope by ZEISS Sigma 300 (Zeiss, Oberkochen, Germany).

#### 2.4.2. Physiological and Biochemical Characteristics

The physiological and biochemical reactions of antagonistic strains were tested according to Fang’s (1998) method [27].

#### 2.4.3. Molecular Identification of Antagonistic Strains

The antagonistic strains were inoculated in NA liquid medium and then cultured at 37 °C, 180 rpm for 24 h. Bacterial genomic DNA was extracted according to the Lysis Buffer for Microorganism to Direct PCR method (TaKaRa, Dalian, China). The partial sequence of 16S rRNA gene was amplified by PCR with universal primers 27F/1492R (5’-AGAGTTTGATCCTGGCTCAG-3’/5’-GGTTACCTTGTTACGACTT-3’) [28]; the primer sequence were synthesized by Shanghai Personal Biotechnology Co., Ltd. (Shanghai, China). The PCR products were sent to Shanghai Personal Biotechnology Co., Ltd. for sequencing. The amplification system (25 μL) comprised: 22 μL of Golden Star T6 super PCR mix (TsingKe, Beijing, China), 1 μL of template DNA (concentration: 50–500 ng), 1 μL of upstream primer, and 1 μL downstream primer. The PCR products were sequenced, and the sequencing results were compared by blast on the NCBI website (https://www.ncbi.nlm.nih.gov/). The phylogenetic tree was constructed by the adjacency method using MEGA 7.0 software (https://www.megasoftware.net/, accessed on 23 February 2021).

### 2.5. Biological Characteristics of Antagonistic Strains

#### 2.5.1. Effect of Temperature on Growth of Antagonistic Strains

The antagonistic strains were activated at 37 °C for 24 h, and then the strains were inoculated in LB medium by shake culturing for 16 h, at 180 rpm. Inoculation was carried out on 50 μL bacteria suspension in a 5 mL LB medium at a concentration of 1% (*v*/*v*), which was then placed, respectively, at temperatures of 25, 28, 31, 34, 40 and 43 °C, in a shaking incubator, and finally they were cultured at 180 rpm for 12 h. An ultraviolet (UV) spectrophotometer was used to measure the absorbance value at 600 nm wavelength, and the average OD_600_ value was calculated. Each treatment was replicated three times.

#### 2.5.2. Effect of pH Value on Growth of Antagonistic Strains

The pH values of LB medium were adjusted to the range of 4.0, 5.0, 6.0, 7.0, 8.0, 9.0 and 10.0, respectively, using 1 mol/L hydrochloric acid solution and 1 mol/L sodium hydroxide solution. The activated strains were inoculated in LB medium by culturing at 37 °C, 180 rpm for 16 h. Inoculating 50 μL bacteria suspension into a 5 mL LB medium with the different pH values at a concentration of 1% (*v*/*v*), then incubating at 37 °C, 180 rpm for 12 h. UV spectrophotometry was used to measure the absorbance value at 600 nm wavelength, and the average OD_600_ value calculated. Each treatment was replicated three times.

### 2.6. Greenhouse Pot Experiment

#### 2.6.1. Pot Experiment Plantlets

Greenhouse pot experiment was carried out from May to October 2020. The tissue cultured plantlets of the Brazilian banana cultivar were transferred into sand substrate to acclimatize for 30 days. The plantlets were then transplanted into plastic pots (11 cm (diameter) × 12 cm (height)) filled with banana planting substrate produced by Yunnan Yuxi Leshi Technology Co., Ltd. (Yunnan China). Banana plantlets with 5–6 leaves were selected to conduct the pot experiment to assess TR4 biocontrol and growth-promoting effects.

#### 2.6.2. Preparation of Fermentation Broth

The antagonistic *Bacillus* strains were activated and cultured at 37 °C for 24 h; single colonies were picked using an aseptic inoculation loop and inoculated in LB medium. The fermentation broth was obtained after shaking culture at 37 °C and 220 rpm for 48 h. Sterile water was used to dilute the suspension to 1 × 10^8^ cfu/mL solution and 1 × 10^7^ cfu/mL fermentation solution.

#### 2.6.3. Preparing the TR4 Spore Suspension

After the TR4 strain 15-1 was activated and cultured at 28 °C for 7 days on PDA, the activated hyphae were placed in PDB medium and culture was shaken at 28 °C and 180 rpm for 72 h. Then, four layers of sterile gauze were used to filter the hyphae to obtain the TR4 spore suspension. Sterile water was used to dilute the suspension to 1 × 10^6^ cfu/mL spore solution of TR4.

#### 2.6.4. Pot Experiment

The fermentation broth (1 × 10^8^ cfu/mL, labeled as “DC1”) and 10-time diluted fermentation broth (1 × 10^7^ cfu/mL, labeled as “DC2”) were used to drench the roots of vigorous banana plants at 40 mL per plant. After 7 days, 1 × 10^6^ spores/mL TR4 spore suspension (labeled as “TR4”) was applied to the banana plant roots and each plant was drenched with 40 mL. The LB culture medium without inoculated strains was used as control (CK) (Table 1). Treatments I, II and III were inoculated with TR4, while treatments IV, V and VI were not inoculated with TR4. Each treatment was with three replicates and 4 bananas plantlets in each replicate.

#### 2.6.5. Investigation of TR4 Biocontrol Effects

The biocontrol effects of *Bacillus* strains on TR4 in pot experiment were investigated in treatments I, II, and III, which were inoculated with TR4. Forty days after inoculation with TR4, the leaf and corm symptoms of the banana plantlets were investigated. Disease severity index (DSI) and control effect were calculated according to the literature [6,29,30].

Disease severity index = ∑ (number of diseased plants at all levels × representative value of the level)/(total number of plants × representative value of the highest level) × 100.

Control effect (%) = (control disease index − treatment disease index)/control disease index × 100.

#### 2.6.6. Evaluation of Growth-Promoting Effects of Biocontrol Bacillus Strains on Banana Plantlets

The growth-promoting effects of biocontrol strains were investigated three times: on the initial day of inoculating antagonistic bacteria, then 20 and 40 days after inoculation with TR4. The plant height, pseudostem girth and leaf number of each banana plantlet in 6 treatments were measured and recorded every time.

Plant height was measured from the ground to the intersection of the two petioles at the top by using tape measure; pseudostem girth was measured from the base of pseudostem, which was 1 cm above the ground by using vernier calipers; number of leaves was recorded as the number of green leaves per plantlet.

### 2.7. PCR Amplification of Biocontrol and Plant Growth-Promotion Related Genes in YN0904 and YN1419

The strains preserved in glycerin (50% sterilized glycerin mixed with equal volume of bacterial suspension) at −80 °C refrigerator were inoculated on NA medium plate by the streak-cultivation method and cultured for 14–16 h at 37 °C. Lysis Buffer for Microorganism to Direct PCR (Takara, Dalian, China) kit was used to prepare the genomic DNA of the *Bacillus*. The activated single colony was picked up with sterilized toothpicks and then stirred in a sterile microtube contained with 50 μL Lysis Buffer and removed. After 15 min of 80 °C thermal denaturation, centrifugation was performed at low speed, and 1–5 μL of supernatant lysate was used as follow-up PCR template.

The genomic DNA of YN0904 and YN1419 was amplified by PCR using twelve pairs of primers which target *Bacillus*’s twelve biocontrol genes, as shown in Appendix A and a plant growth-promotion related gene *ysnE* (Primers: 5′-GGCTGTAACCTTTGCTATG-3′ and 5′-GCTGTTCGGGTCCTCTTTAT-3′ with amplicon size: 488 bp). The PCR amplification system (total 25 μL) were 12.5 μL of MegaFTM Fidelity 2X PCR MasterMix (Applied Biological Materials Inc., NanjingChina), 1 μL of upstream primer (10 µM), 1 μL of downstream primer (10 µM), 10.5 μL of Nuclease-free H_2_O and 1 μL of template genomic DNA. The reaction components were run for 98 °C for 30 s, followed by 35 cycles of 98 °C for 10 s, 55–62 °C for 30 s and 72 °C for 30 s, and total extending at 72 °C for 2 min. No-template reaction replacing genomic DNA with Nuclease-free H_2_O was used as negative control. *B. velezensis* strain YN1282-2, which is positive for all the biocontrol genes used in this study, was used as a positive control [21]. The amplified products were detected by agarose gel electrophoresis (1.2 g/100 mL).

### 2.8. Data Analysis

The data were analyzed by SPSS 18.0, Duncan’s new complex range method and independent sample *t* test. The average data are expressed as “means ± standard error (S.E.)”.

## 3. Results

### 3.1. Isolation and Screening of Antagonistic Endophytic Bacteria

Since July 2009, a total of 256 plant samples were collected from different banana-producing areas, including Xishuangbanna, Yuxi, Honghe, Wensahng, Baoshang and Dehong of Yunnan Province. As a result, a total of 813 strains of endophytic bacteria were isolated and 382 of them were selected for antagonism screening against TR4 (Appendix A). After the dual culture for primary screening, nine antagonistic endophytic bacteria were obtained. After the dual-culture for secondary screening, two antagonistic endophytic strains YN0904 and YN1419 with the best antagonistic effect on TR4 were obtained (Appendix A), which were isolated from symptomless pseudostem samples already infected with FWB and bacterial soft rot, and the sampling stage was at banana bud formation. YN0904 was isolated from the *Musa* ABB banana variety group, cv Pisang Awak, originating from Mengla (101°18’12″ E; 21°27’21″ N; altitude 767 m), Xishuangbanna, Yunnan in August 2009. The FWB incidence in this plantation was 3–5%. YN1419 was isolated from the Brazilian banana variety, originating from Daluo (99°60’18″ E; 21°36’51″ N; altitude 598 m), Xishuangbanna, Yunnan, in August 2014. The FWB incidence in this plantation was 6–8%.

### 3.2. Effectiveness of Antagonistic Endophytic Bacteria In Vitro

The two strains with the strongest antagonist activity against TR4 by dual culture of secondary screening were labeled as YN0904 and YN1419 (Figure 1a,b and Appendix A). After being subjected to dual culture with antagonistic bacteria for 7 days, the TR4 hyphal were swollen and deformed, the tops of hyphae were expanded, and the hyphae internodes between diaphragms were shorter (Figure 1d,e). The hyphae of the TR4 control (CK) were normal, smooth and uniform (Figure 1f). The average diameter of TR4 hyphae of YN0904 against TR4 was 1.8 cm, and the inhibition rate was 79.6% (Table 2). The average diameter of TR4 hyphae of YN1419 against TR4 was 1.7 cm, and the inhibition rate was 81.3% (Table 2).

### 3.3. Morphological Characteristics of Antagonistic Strains

The colonies of endophytic bacteria YN0904 had a light yellow color, opaque and rough surface, middle convex and irregular edge on NA medium after 24 h culture at 37 °C (Figure 2a). Further, scan electron microscopy (SEM) showed that the bacteria were rod-shaped and oval at ends and the size of bacteria was 2.3–2.5 μm × 0.5–0.8 μm (Figure 2b). The colonies of endophytic bacteria YN1419 had a grayish-white color, convex, irregular edge, opaque and rough surface on NA after 24 h of culturing at 37 °C (Figure 2c). SEM showed that the bacteria were rod-shaped and oval at ends; the size of bacteria was 2–3 μm × 0.8–1 μm. (Figure 2d).

### 3.4. Physiological and Biochemical Characteristics of Antagonistic Strains

Physiological and biochemical tests of strains YN0904 and YN1419 are shown in Table 3. YN0904 was identified as a Gram-positive bacterium, and showed positive to the test of sucrose, α-D-Glucose, amylose hydrolysis, etc. The isolate YN1419 was also identified as a Gram-positive bacterium, and tested positive for sucrose, α-D-Glucose, amylose hydrolysis, etc. We also determined the optimal temperature and pH for YN0904 and YN1419 growth were 37 °C and 7.0, respectively (Appendix A).

### 3.5. Molecular Identification of Antagonistic Bacteria

When aligning the 16S sequence of antagonistic strain YN0904 (GenBank Accession No. MW647760) in GenBank, the highest similarity sequence was with *B. amyloliquefaciens*. The 16S rRNA gene fragment of YN0904 was 1420 base pairs (bp). The phylogenetic tree based on 16S rRNA gene sequence results showed that strain YN0904 and *B. amyloliquefaciens* (MT613661, 65777, MW082822) clustered in the same large branch with a 69% bootstrap value (Figure 3). As a conclusion, YN0904 was identified as *B. amyloliquefaciens* based on its molecular, morphology, physiological and biochemical characteristics.

When aligning the 16S sequence of the antagonistic strain YN1419 (GenBank Accession No. MW647761) in GenBank, the highest degree of sequence similarity was with B. subtilis. The 16S rRNA gene fragment of YN1419 was 1419 base pairs (bp). The phylogenetic tree based on 16S rRNA gene sequence results showed that this strain and *B. subtilis* (MT538531, MT513998) clustered in the same large branch with a 99% bootstrap value (Figure 3). As a conclusion, YN1419 was identified as *B. subtilis* based on its molecular, morphological, physiological and biochemical characteristics.

### 3.6. YN0904 and YN1419 Had Good Biocontrol Effects on TR4 in Pot Experiment

At 40 days post inoculation (dpi), the leaves of the control group (CK + TR4) that had been inoculated only with TR4 turned yellow and the plantlets were stunted, while most leaves of the treated groups DC1 of both YN0904 and YN1419 remained healthy (Figure 4a). Additionally, after splitting the corm, we found that the most corms of treated groups DC1 and DC2 of both YN0904 and YN1419 were hardly invaded by TR4 hyphae, while TR4 caused severe damage in the control—the color of the corms was red or brownish-black (Figure 4b).

Disease investigation showed that there was a significant TR4 suppressive effect of the two antagonistic bacteria. For the disease indexes, the treated groups DC1 and DC2 of both YN0904 and YN1419 on banana corms and leaves were significantly lower than that in treated groups. Among them, there was no significant difference between YN0904 DC1 and DC2, but YN1419 DC1 was significantly lower than DC2 (Figure 4a,b and Table 4).

For the biocontrol effects, the treated groups DC1 of YN0904 on banana corms and leaves were slightly higher than DC2, respectively, but the difference between them was not significant. The treated groups DC1 of YN1419 on banana corms and leaves were significantly higher than DC2. There was no significant difference between the treated groups DC1 of YN0904 and YN1419 on banana corms and leaves, respectively (Figure 4a,b and Table 4).

### 3.7. Growth-Promoting Effect of Antagonistic Bacteria on Banana Plantlets

The inoculation of two biocontrol strains YN0904 and YN1419 did not affect the growth of banana plantlets, and YN0904 also had obvious growth-promoting effects on banana plants (Figure 4c).

For YN0904, at 20 days post inoculation (dpi), there was no significant difference in plant height (DC1: 28.99 ± 2.39 cm; DC2: 25.07 ± 1.18 cm), pseudostem girth (DC1: 9.15 ± 0.51 mm; DC2: 9.40 ± 0.54 mm) and leaf number (DC1: 6.75 ± 0.30; DC2: 6.08 ± 0.23) with CK (plant height: 27.12 ± 1.13 cm; pseudostem girth: 9.25 ± 0.25 mm; leaf number: 6.58 ± 0.26), respectively. At 40 dpi, the plant height and pseudostem girth of YN0904 DC1 were 36.17 ± 1.10 cm and 11.64 ± 0.48 mm, which were significantly higher than those of CK (30.58 ± 1.24 cm and 9.88 ± 0.21 mm), respectively. Additionally, there was no significant difference in plant height and pseudostem girth of YN0904 DC2 (31.44 ± 0.40 cm and 10.04 ± 0.47 mm) with CK. For leaf number, there was no significant difference of DC1 (20 dpi: 6.75 ± 0.30; 40 dpi: 7.17 ± 0.39) and DC2 (20 dpi: 6.08 ± 0.23; 40 dpi: 6.33 ± 0.19) with CK at 20 (6.58 ± 0.26) and 40 dpi (6.75 ± 0.18), respectively (Figure 5a, Figure 6a and Figure 7a).

For YN1419, there was no significant difference in plant height (20 dpi: DC1: 28.75 ± 1.39 cm; DC2: 29.00 ±2.25 cm; 40 dpi: DC1: 32.39 ± 1.72 cm; DC2: 32.10 ± 2.51 cm), pseudostem girth (20 dpi: DC1: 8.54 ± 0.52 mm; DC2: 8.88 ± 0.53 mm; 40 dpi: DC1:10.68 ± 0.52 mm; DC2: 9.85 ± 0.60 mm) and leaf number (20 dpi: DC1: 7.50 ± 0.34; DC2: 7.00 ± 0.48; 40 dpi: DC1: 7.75 ± 0.39; DC2: 7.42 ± 0.31), with CK at 20 and 40 dpi, respectively (Figure 5b, Figure 6b and Figure 7b).

### 3.8. PCR Amplification Results of Biocontrol and Plant Growth-Promotion Related Genes in YN0904 and YN1419

For twelve biocontrol related genes, seven non-ribosomal peptide synthetases (NRPS) genes including *fenD*, *ituC*, *yngG*, *yndJ*, *srfAA, bamD*, and *dhb*, which are responsible for the syntheses of seven non-ribosomal metabolites, were presented in YN0904, except for *srfAA* (Table 5 and Appendix A). Among the four polytide synthetases (PKS) genes, including *dfn*, *bae*, *mln* and *bac*, which were responsible for the syntheses of four polyketides metabolites, YN0904 contains *dfn*, *bae* and *bac* genes (Table 5 and Appendix A). Surprisingly, there are no NRPS and PKS genes in the YN1419 genome (Table 5 and Appendix A). Only the marker gene *sboA* was detected in YN1419, while it was not detected in YN0904 (Table 5 and Appendix A).

The plant growth-promotion related gene *ysnE*, which is responsible for the synthesis of auxin, was only detected in YN0904 (Table 5 and Appendix A).

## 4. Discussion

The chemical components of some fungicides have a disinfecting effect on TR4 in vitro, but the control effect in soil environment is not clear, and it is unfriendly to the environment [31]. At present, chemical control measures have no significant effect on the control of FWB in the field, and biological control is the most potential and effective method [1]. Endophytic bacteria are a kind of potential biocontrol resource, which can secrete a variety of antibacterial substances and reproduce independently in plants [32,33]. In this study, further antagonistic determination of the isolated strains was conducted; the endophytic bacterial strains YN0904 and YN1419 showed the TR4 antagonistic effect in vitro (Figure 1 and Appendix A), which also showed significant suppressive effects on TR4 in vivo (Figure 4a,b and Table 4). These two strains have no deleterious effects on banana growth and YN0904 also has an obvious growth-promoting effect (Figure 4c, Figure 5, Figure 6 and Figure 7). YN0904 could produce a variety of antibiotics to inhibit the growth of TR4, which could alleviate the symptoms of plants. The strain YN0904 contains IAA synthesis genes (Table 5), which would synthesize IAA and promote to the normal growth of plants after being infected by TR4. Surprisingly, the marker gene *sboA* could only be detected in YN1419 while the other 11 marker genes are all absent (Appendix A). Subtilosin A was proved as a broad-spectrum antibiotic against bacterial pathogens, but its anti-fungal effect has not been reported. In addition, the *ysnE* gene could not be detected in the YN1419 genome (Appendix A). YN1419 may antagonize TR4 and promote plant growth through the synthesis of new substances or through new biocontrol mechanisms.

By morphological, molecular identification, physiological and biochemical characteristics, two strains, YN0904 and YN1419, were identified as *B. amyloliquefaciens* and *B. subtillis*, respectively. *Bacillus* is a family of bacteria with rich diversity, which has many qualified characteristics such as stress resistance, broad-spectrum antibacterial, environmental friendly, low pathogenicity to plants, and resilient adaptability properties [34]. The biocontrol *Bacillus* could produce abundant secondary metabolites during the growth process to inhibit the activity of pathogenic fungi or bacteria and have a promoting effect on plants growth [35]. They are considered to be ideal biocontrol agent against soil-borne diseases such as FWB [36,37,38].

At present, research on biocontrol of FWB are mainly focused on morphology assays or pot experiments; the efforts to explore the antagonistic mechanisms of biocontrol strains are limited. *B. amyloliquefaciens* is a kind of main bacterium used in biological control of FWB [39,40,41]. In this study, the control effect of endophytic *B. amyloliquefaciens* YN0904 on FWB was investigated both on leaves and corm of banana plants. The control effect of YN0904 is better than other endophytic *B. amyloliquefaciens* reported in previous studies. Zhu et al. [42] have reported that endophytic *B. amyloliquefaciens* had good control effects against Fusarium chlamydospore. Marcos et al. [43] reported an endophytic *B. amyloliquefaciens* played a functional role in enhancing growth and disease protection of invasive English ivy (Hedera helix). *B. subtillis* is a similar species to *B. amyloliquefacien*. It usually has a broad-spectrum antibacterial activity and growth-promoting effects. It is widely used in biological control of plant diseases in agricultural production [44,45]. There are few reports about the control effect of *B. subtilis* on FWB. In this study, the control effect of endophytic bacteria *B. subtilis* YN1419 on FWB was obvious.

Previous studies have shown that in the lipopeptide metabolites synthesized by *Bacillus*, iturin and fengycin had strong antagonistic activity against pathogenic fungi, and there was a synergistic effect between iturin and fengycin [46,47]. Bacillomycin D and fengycin were related to the antagonistic effect of *B. amyloliquefaciens* SQR9 on TR4 [48]. The crude extract of BEB17 contains bacillibactin, difficidin and bacillaene, etc., which had a good inhibitory effect on TR4 [49]. Li and He et al. detected 12 biocontrol marker genes by PCR in five *Bacillus* strains with good TR4 antagonistic effects and found that most of the biocontrol genes could be detected in these five strains, and they speculated that the antibacterial effect of the five *Bacillus* strains might be due to the presence of key biosynthesis genes of these biocontrol metabolites in their genomes [21]. In this study, most of the biocontrol marker genes were detected in the genomes of YN0904, except for *srfAA*, *mln* and *sboA* (Table 5 and Appendix A). We speculated that the biocontrol effect of YN0904 on TR4 may be related to the synthesis of these substances. Surprisingly, only the *sboA* gene was detected in the YN1419 genome, even its biocontrol effect on TR4 was slightly better than that of YN0904. However, the corresponding antibiotic of *sboA* is subtilosin A, which is a broad-spectrum antibiotic against bacterial pathogens, but its resistance to fungi has not been reported so far [50]. Therefore, it is likely that there are some new antagonistic substances or new antagonistic mechanisms in YN1419, but it needs further confirmation in our next step of research. In addition, *ysnE*, a key IAA synthase gene [25], was detected in YN0904 (Table 5 and Appendix A), indicating that the plant growth-promoting effect of YN0904 when plant was infected with TR4 may be related to the biosynthesis of auxin. However, the *ysnE* gene is not present in the YN1419 genome, but it has a significant growth-promoting effect on TR4 infected banana. We will sequence the whole genome of YN1419 next to further explore the TR4 biocontrol and banana growth-promoting mechanism of YN1419, and the specific synthesis genes of these biocontrol and growth-promoting substances need to be further verified by specific gene knock-out in the next step of our research.

The banana industry offers a key route to poverty alleviation and increasing local farmers’ incomes in remote areas of Yunnan for ethnic groups. However, the current banana industry is seriously threatened by TR4. After years of screening antagonistic bacteria against FWB, two strongly antagonistic strains belonging to endophytic bacteria have now been identified from local banana plantations. The mechanisms for disease suppression by antagonistic strains YN0904 and YN1419, and monitoring their field-level control effects need to be further studied and validated. Furthermore, these two natively strains could be developed into biofertilizer in subsequent steps of our research because they are already adapted to local ecological conditions, which is a promising way to solve the problem and contributes to the stable development of the banana industry.

## 5. Conclusions

Two TR4 antagonistic endophytic bacteria were identified from banana plants, which were identified as *B. amyloliquefaciens* and *B. subtilis* by morphological and molecular biology and labeled as YN0904 and YN1419, respectively. The pot experiment showed that YN0904 and YN1419 had significant inhibitory effects on FWB, and these two strains had no significant deleterious plant growth effect on banana growth. The PCR results showed that several biocontrol marker genes related to the biosynthesis of antibiotics synthesized by NRPS, PKS and RPS pathways and IAA key synthetase gene could be detected in the genomes of YN0904. However, the genomes of YN1419 contain only one biocontrol gene *sboA*, which is a key synthase gene of subtilosin A. Molecular characterization provided a theoretical basis for expounding the biocontrol mechanisms of these two strains.

## Figures and Tables

**Figure 1 jof-07-00795-f001:**
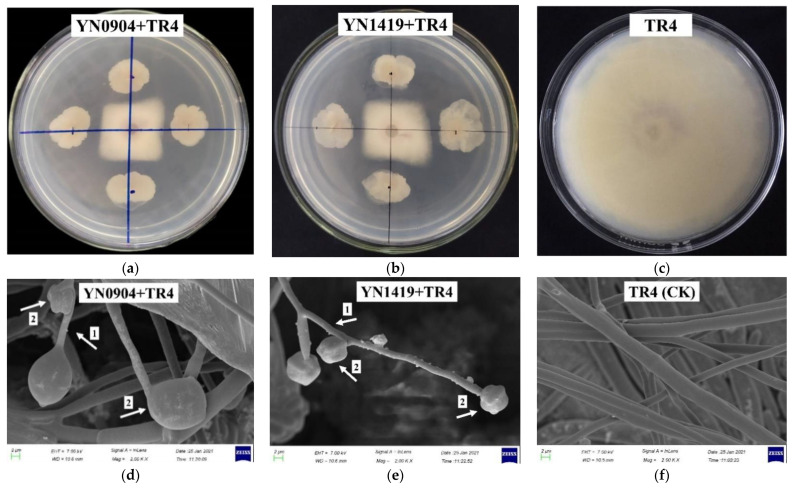
The antagonistic effect of strains YN0904 and YN1419 on TR4. (**a**) The antagonistic effect of strain YN0904 on TR4; (**b**) the antagonistic effect of strain YN1419 on TR4; (**c**) control of TR4; (**d**) characteristics of hyphae morphology of strain YN0904 antagonistic to TR4 (Arrow 1: The internodes of TR4 hyphae were shortened; Arrow 2: The top of the TR4 hyphae expanded); (**e**) characteristics of hyphae morphology of strain YN1419 antagonistic to TR4 (Arrow 1: The internodes of TR4 hyphae were shortened. Arrow 2: The top of the TR4 hyphae expanded); (**f**): characteristics of hyphae morphology of control TR4.

**Figure 2 jof-07-00795-f002:**
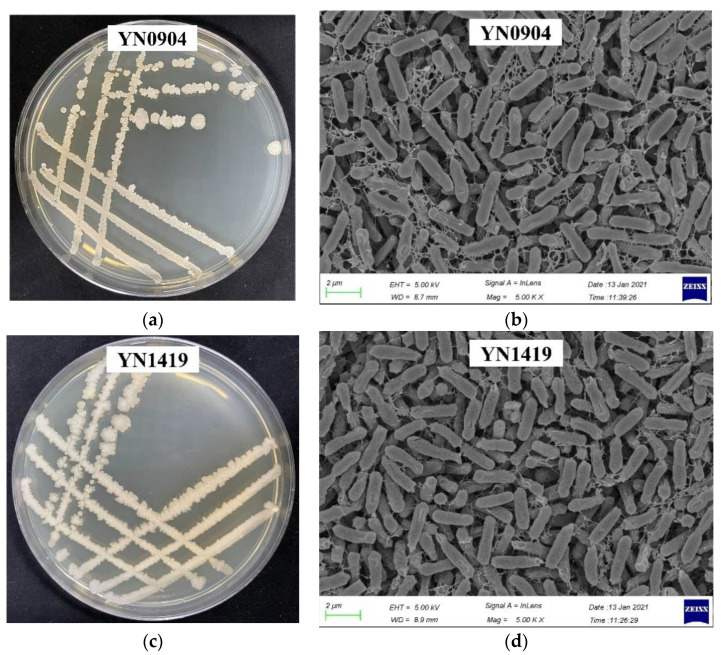
Colony and morphology of the antagonistic strains YN0904 and YN1419 cultured on NA medium at 30 °C for 24 h. (**a**) Colony morphology of the antagonistic strains YN0904. (**b**) Scanning electron micrograph of the antagonistic strains YN0904. (**c**) Colony morphology of the antagonistic strains YN1419. (**d**) Scanning electron micrograph of the antagonistic strains YN1419.

**Figure 3 jof-07-00795-f003:**
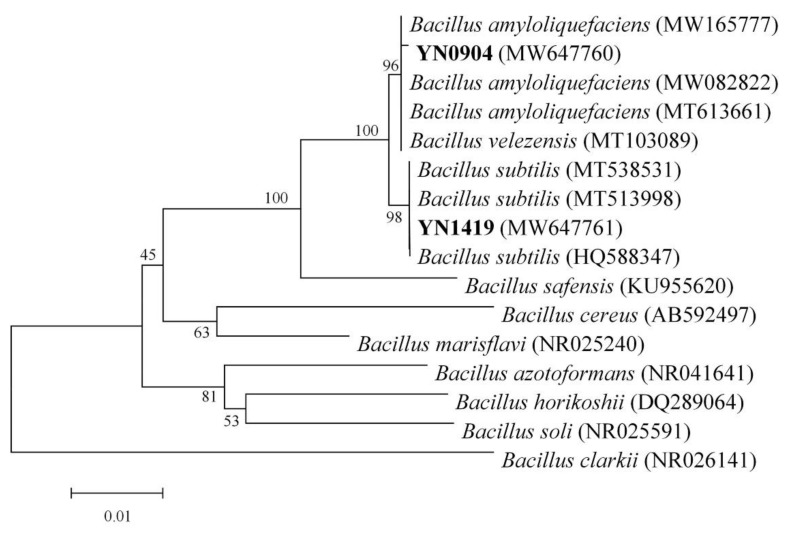
Phylogenetic tree based on 16S rRNA gene sequences of the antagonistic strain YN0904 and YN1419.

**Figure 4 jof-07-00795-f004:**
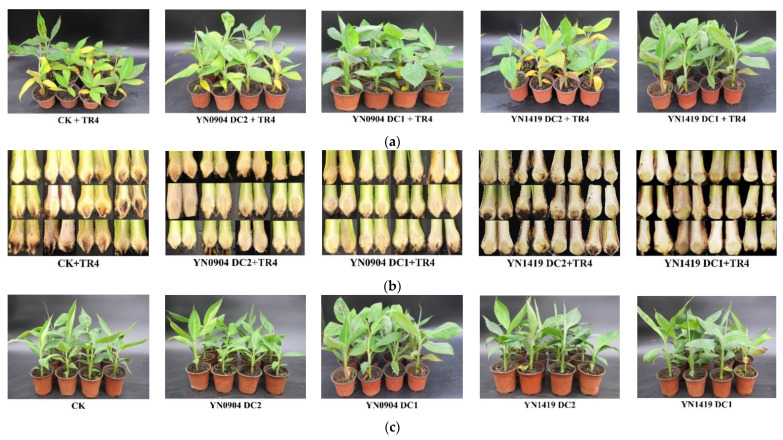
Biocontrol effects and growth-promoting of antagonistic strains YN0904 and YN1419. (**a**) Control effect of antagonistic strains YN0904 and YN1419 against TR4 in leaves; (**b**) control effect of antagonistic strains YN0904 and YN1419 against TR4 in the corm; (**c**) growth-promoting effect of antagonistic strains YN0904 and YN1419.

**Figure 5 jof-07-00795-f005:**
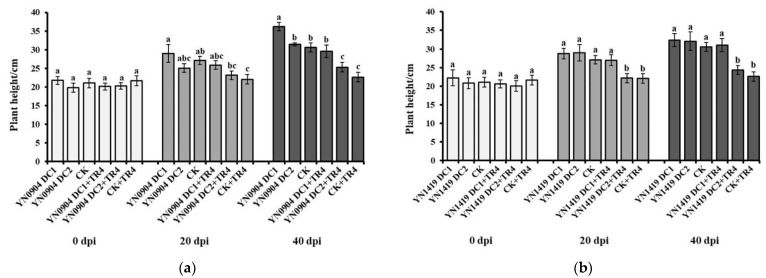
Growth-promoting effects of antagonistic strains YN0904 and YN1419 on banana plant height. (**a**) Growth-promoting effects of antagonistic strain YN0904 on banana plant height. (**b**) Growth-promoting effects of antagonistic strain YN1419 on banana plant height. Data are presented as means ± standard error. Data with different lowercase letters indicate a significant difference at the 0.05 level.

**Figure 6 jof-07-00795-f006:**
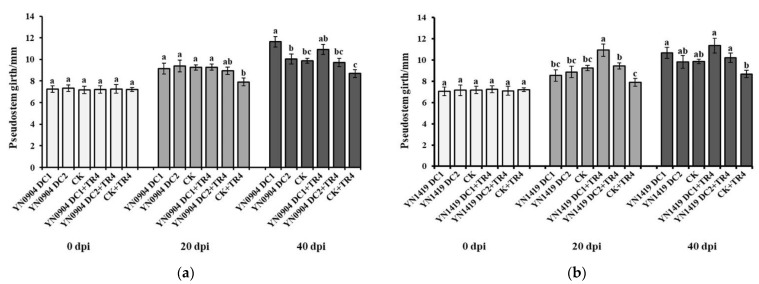
Growth-promoting effects of antagonistic strains YN0904 and YN1419 on pseudostem girth. (**a**) Growth-promoting effects of antagonistic strain YN0904 on pseudostem girth; (**b**) growth-promoting effects of antagonistic strain YN1419 on pseudostem girth. Data are presented as means ± standard error. Data with different lowercase letters indicate a significant difference at the 0.05 level.

**Figure 7 jof-07-00795-f007:**
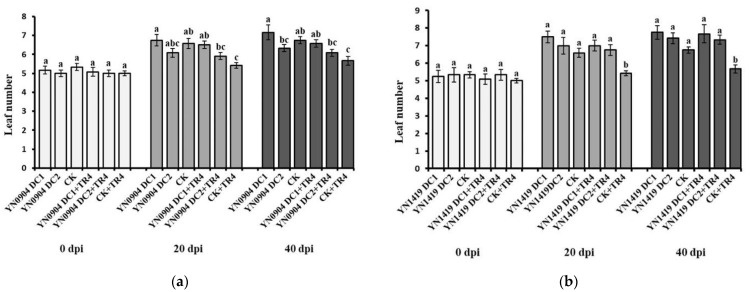
Growth-promoting effects of antagonistic strains YN0904 and YN1419 on leaf number. (**a**) Growth-promoting effects of antagonistic strain YN0904 on leaf number; (**b**) growth-promoting effects of antagonistic strain YN1419 on leaf number. Data are presented as means ± standard error. Data with different lowercase letters indicate a significant difference at the 0.05 level.

**Table 1 jof-07-00795-t001:** Different treatments in pot experiment with *Bacillus* strains YN0904 and YN1419.

Group	Treatment
I	Inoculation of fermentation broth + spore suspension of TR4 (DC1 + TR4)
II	Inoculation of 10 times diluted fermentation broth + spore suspension of TR4 (DC2 + TR4)
III	Inoculation of blank LB liquid medium+ spore suspension of TR4 (CK + TR4)
IV	Inoculation of fermentation broth only (DC1)
V	Inoculation of 10 times diluted fermentation broth only (DC2)
VI	Inoculation of blank LB liquid medium only (CK)

**Table 2 jof-07-00795-t002:** Antagonistic activities of stains YN0904 and YN1419 against TR4.

Strains	Diameter of TR4 Colony (cm)	Inhibition Rate (%)
YN0904	1.8 ± 0.17 b	79.6 ± 0.11 **
YN1419	1.7 ± 0.17 c	81.3 ± 0.22 **
CK	9.0 ± 0.03 a	/

Data are presented as means ± SD. Data with different lowercase letters indicate a significant difference at the 0.05 level. ** indicates a significant difference at the 0.01 level.

**Table 3 jof-07-00795-t003:** Physiological and biochemical characteristics of the antagonistic strains YN0904 and YN1419.

Item Tested	Reaction		Item Tested	Reaction	
	YN0904	YN1419		YN0904	YN1419
Gram stain	+	+	D-Sorbitol	+	+
Sucrose	+	+	D-Mannitol	+	+
α-D-Glucose	+	+	1% NaCl	+	+
D-Maltose	+	+	4% NaCl	+	+
Fructose	+	+	8% NaCl	+	+
Mannose	+	+	Amylose hydrolysis	+	+

“+” represents bacteria positive for physiological and biochemical characteristics.

**Table 4 jof-07-00795-t004:** Biocontrol effects of the antagonistic strains YN0904 and YN1419 on the TR4 in pot experiment.

Treatment	Disease Index		Control Effect (%)	
	Corm	Leaf	Corm	Leaf
YN0904 DC1 + TR4	14.58 ± 2.08 c	12.50 ± 3.61 c	74.26 ± 2.27 a	82.58 ± 4.61 a
YN0904 DC2 + TR4	18.75 ± 3.61 c	20.83 ± 2.08 bc	67.22 ± 4.33 a	70.71 ± 2.02 ab
YN1419 DC1 + TR4	16.67 ± 2.08 c	10.42 ± 4.17 c	70.09 ± 4.41 a	85.61 ± 5.30 a
YN1419 DC2 + TR4	37.50 ± 6.25 b	31.25 ± 6.25 b	32.87 ± 11.46 b	56.30 ± 7.34 b
CK + TR4	56.25 ± 3.61 a	70.83 ± 2.08 a		

Data are presented as means ± standard error. Data with different lowercase letters indicate a significant difference at the 0.05 level.

**Table 5 jof-07-00795-t005:** The overview of biocontrol marker gene and synthetic gene amplification status in tested strains of *Bacillus* spp. YN0904 and YN1419.

Category	Metabolites	Synthesis Gene	YN0904	YN1419
Non-ribosomal peptide synthetases (NRPS)	Surfactin	*srfAA*	−	−
	Fengycin	*fenD*	+	−
	Iturin	*ituC*	+	−
	bacillomycine D	*bamD*	+	−
	yngG	*yngG*	+	−
	yngJ	*yndJ*	+	−
	Bacillibactin	*dhb*	+	−
Polytide synthetases (PKS)	Difficidin	*dfn*	+	−
	Bacillaene	*bae*	+	−
	Macrolactin	*mln*	−	−
	Bacilysin	*bac*	+	−
Ribosomal peptide synthetases (RPS)	Subtilosin	*sboA*	−	+
	Auxin	*ysnE*	+	−

The genomic DNA of *Bacillus* strains YN10904 and YN1419 was amplified respectively by PCR using thirteen pairs of published primers which target twelve biocontrol marker genes and a plant growth-promotion gene *ysnE*. “+” represents the positive result of PCR, “−” represents the negative result of PCR.

## Data Availability

Two strains, YN0904 and YN1419, were registered in NCBI, GenBank Accession No. of YN0904 is MW647760, GenBank Accession No. of YN1419 is MW647761.

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
