# Peer review of "Biological Control of Fusarium oxysporum f. sp. cubense Tropical Race 4 Using Natively Isolated Bacillus spp. YN0904 and YN1419"

_jof, 2021, doi:10.3390/jof7100795_

Round 1
Reviewer 1 Report
Dear Authors.
The manuscript described research related to one of the most problematic phytopathogen worldwide, such as Fusarium oxysporum f. sp. cubense. The results showed a promissory inhibition using biological control, becoming a relevant alternative for the agricultural field. However, I suggest you include a comparison against chemical control using commercial and representative agrochemical agents. I guess this experimental comparison could demonstrate the relevance of your biological control, improving the manuscript's impact.
Author Response
Response to Reviewer 1 Comments
We were pleased to read the insightful evaluations of the reviewer 1.
The manuscript described research related to one of the most problematic phytopathogen worldwide, such as Fusarium oxysporum f. sp. cubense. The results showed a promissory inhibition using biological control, becoming a relevant alternative for the agricultural field.
Response 1: Thank reviewer 1 for this positive recommendation.
Point 2: However, I suggest you include a comparison against chemical control using commercial and representative agrochemical agents. I guess this experimental comparison could demonstrate the relevance of your biological control, improving the manuscript's impact.
Response 2: Thank reviewer 1 for this kind of suggestion. Relevant literature indicates that the chemical components of some fungicides have disinfecting effect on TR4 in vitro, but the control effect in soil environment is not clear, and it is unfriendly to the environment. At present, chemical control measures have no effective effect on the control of FWB in the field, and biological control is the most potential and effective method. According to your suggestion, we have added relevant contents in the discussion part of the manuscript, see line 469-472.
Sincerely and with our best regards,
Si-Jun Zheng on behalf of the co-authors

Reviewer 2 Report
The revised manuscript has made significant improvement compared to the previous version. The reviewer believe it is acceptable after satisfactorily addressing the following comments:
- As pointed in the comments to the original manuscript on the proper decimal points to use to express the accuracy of the OD readings, which the authors have addressed. However, your response to my earlier comment did not address my question. You can read your culture under OD600 for a min, and the reading will keep on changing. If you take the reading at two different time, let’s say in 10 sec apart, the last two digits in the reading are most likely different. Therefore, your data can not be accurate to the fourth decimal point although you can read the fourth decimal point. This is a fact. The correct way of reporting the analyzed data is you only report to the first digit that is not reliable. The same principle is true for the inhibition rate (such as 79.55%). I have not seen any one conducting a biological study and can reach an accuracy of 0.01%. The highest I have only seen is 0.1% accuracy in analytical chemistry among the technical repeats. It is meaningless to report data including two digits of values that are estimated (due to averaging you lose one more digit of accuracy). Please also see my comment on line 277 and correct throughout the manuscript.
- Please combine Fig 3 and Fig 4 into one tree.
- Also, please move section 3.6 and 3.7 into supplement due to the fact the experiment was not conducted properly. The growth rate through OD reading should be measured during the logarithmic phase of the bacterial growth, which is usually within 4-6 hrs. When the OD600 is reaching over 2 after 12 hr, the reading has long passed its linear accuracy range of the spectrophotometer. The bacteria are already in the stationary phase, way past the log phase. The difference of effects between different temperature was artificially shrunk due your delay in measurement. As a result, your data accuracy and reliability can not be guaranteed.
- Line 199 of the returned manuscript, I believe for TR4, you do not do colony count, you do spore count. If this true, it should be expressed as cfu/ml.
- Please look over the returned pdf file for additional corrections to incorporate during revision.

Author Response
Response to Reviewer 2 Comments
Point 1: The revised manuscript has made significant improvement compared to the previous version. The reviewer believe it is acceptable after satisfactorily addressing the following comments:
Response 1: We were pleased to read the insightful and overall evaluations of the reviewer 2. Thanks the reviewer 2 for his/her positive recommendation.
Point 2: As pointed in the comments to the original manuscript on the proper decimal points to use to express the accuracy of the OD readings, which the authors have addressed. However, your response to my earlier comment did not address my question. You can read your culture under OD600 for a min, and the reading will keep on changing. If you take the reading at two different time, let’s say in 10 sec apart, the last two digits in the reading are most likely different. Therefore, your data can not be accurate to the fourth decimal point although you can read the fourth decimal point. This is a fact. The correct way of reporting the analyzed data is you only report to the first digit that is not reliable. The same principle is true for the inhibition rate (such as 79.55%). I have not seen any one conducting a biological study and can reach an accuracy of 0.01%. The highest I have only seen is 0.1% accuracy in analytical chemistry among the technical repeats. It is meaningless to report data including two digits of values that are estimated (due to averaging you lose one more digit of accuracy). Please also see my comment on line 277 and correct throughout the manuscript.
Response 2: Thank reviewer 2 for this kind of suggestion. Your comments are very valuable. As you suggested, although we can read the fourth decimal point, we also found that when we read our culture under OD600 at different time, the last 2 digits in the reading are different. We have reorganized the data, and kept the first digit that is not reliable. According to your comment on line 277, we have corrected all the report data.
Point 3: Please combine Fig 3 and Fig 4 into one tree.
Response 3: Thank reviewer 2 for this kind of suggestion. We have combined Fig 3 and Fig 4 into one figure as one tree.
Point 4: Also, please move section 3.6 and 3.7 into supplement due to the fact the experiment was not conducted properly. The growth rate through OD reading should be measured during the logarithmic phase of the bacterial growth, which is usually within 4-6 hrs. When the OD600 is reaching over 2 after 12 hr, the reading has long passed its linear accuracy range of the spectrophotometer. The bacteria are already in the stationary phase, way past the log phase. The difference of effects between different temperature was artificially shrunk due your delay in measurement. As a result, your data accuracy and reliability cannot be guaranteed.
Response 4: Thank reviewer 2 for this kind of suggestion. We have moved section 3.6 and 3.7 into supplement materials as Figure S1 and S2. The effects of temperature and pH on growth of antagonistic strains had no significant effect on the overall results of strains identification.
Point 5: Line 199 of the returned manuscript, I believe for TR4, you do not do colony count, you do spore count. If this true, it should be expressed as cfu/mL.
Response 5: Thank reviewer 2 for this suggestion. For TR4, we indeed do spore count, and we expressed as spore/mL.
Point 6: Please look over the returned pdf file for additional corrections to incorporate during revision.
Response 6: Thank reviewer 2 for this suggestion. We have carefully read the entire pdf file and modified it one by one according to your comments.
Sincerely and with our best regards,
Si-Jun Zheng on behalf of the co-authors

This manuscript is a resubmission of an earlier submission. The following is a list of the peer review reports and author responses from that submission.
Round 1
Reviewer 1 Report
The authors presented a manuscript with detailed methodological information and promising results. The results obtained are of interest since they impact one of the most important commercial and agricultural crops such as Fusarium oxysporum. Furthermore, the authors presented an alternative biological control method that can be implemented in the long term given the continuity of research in this area. Respectfully, I suggest to review some writing and grammar aspects in the manuscript to improve the current version. One of the results that are discussed is the disease severity index, which depends on "the number of diseased plants at all levels". Respectfully, I suggest defining this term within the methodological part, as a scale of symptoms, which could define it better. This will allow a less subjective approach, since the value of this term may depend on the appreciation of the experimenter.
Reviewer 2 Report
This manuscript on the isolation and identification of two Bacillus strains displaying antagonistic effects upon the growth of Fusarium oxysporum f sp cubense and infection of banana plants is a very interesting paper and the experiments are well performed and clearly described. This would be a good paper for a microbiology journal but the lack of development of the effects on the Fusarium questions its suitability for the Journal of Fungi. I suspect from the references quoted in the bibliography that this is part of a series of papers the authors wish to publish on this subject. However, the authors should consider the significant volume of research that has been published on this subject since 2020, and how researchers are identifying why Fusarium infection is being influenced by microbial biocontrol agents. This background information is missing in this submitted manuscript.
Your microscopic study on the effect of the two Bacillus strains on the mycelial morphology is very interesting (Figure 1), and it would be of more impact for the Journal of Fungi is this was followed more closely.
Be careful in what you describe as inhibition in your characterization experiments, as growth in figure 5 was not inhibited at 25oC and 28oC (line 360), but was lower than at the higher temperatures. Similarly with the pH, I think you are meaning below pH 4 and above pH 9 you see inhibition, not as you have written (line 370-371).
Additionally, you selected the only 2 strains that were antagonistic to the growth of Fusarium from your screening, not the "best strains" (line 273).
Reviewer 3 Report
Reviewer’s comments on JOF 1177463:
The manuscript described the discovery of two Bacillus bacterial species that were isolated from banana pseudostems and soil samples, respectively. The manuscript itself is generally well written. However, it is not acceptable in the current form for the reasons listed below:
- The research presented here lacks the depth. It did not provide any new information than what we already know from literature regarding the two species and their potential as biocontrol agents. There are two areas that the authors can do to strengthen this manuscript: conduct the same study under field conditions and report their effectiveness in reducing FWB with a larger samples size (four plantlets per treatment is really not enough); and/or determine the mechanism of antagonism towards Fusarium through isolation and quantification of the secondary metabolite that has been reported in the same species in the earlier studies to be involved in the antagonistic effects or new secondary metabolites.
- The potential of YN0904 and YN1910 as biocontrol agents for managing FWB is over stated since they are not widely present in the collected pseudostem or soil samples, indicating their lack of fitness in the environments. Introducing a non-native and non-adapted strains of bacteria or fungi as a biocontrol agent was never a good idea. This can be argued if the authors can demonstrate the applied bacteria can survive over an extended time in the environment where banana orchards are using the fluorescent strains that the authors mentioned in the discussion.
- Another key weakness of this study is that the authors only examined inoculation 7 days after applying the bacteria, why not examining any other time frame, such as adding both TR4 and the bacteria on the same day or inoculating the TR4 before adding the bacteria???????
- The authors stated that the experiment was repeated three times. However, there was not information on whether the data presented in tables or figures were from a single experiments or combined from three replicated experiments. In addition, the data presented were showing their calculated control effect with accuracy of 0.01%, which I don’t believe this is possible. There are other areas where the authors inflated their data accuracy, for example for OD600, which they presented in the text is a single reading instead of an average since they showed four digits after decimal. An average OD reading of a bacterial culture can not reach an accuracy of 0.0001. Please see my comments in the returned edited pdf file.
- When you dilute the bacterial culture 10 times, it is traditionally labelled as 0.1x, not 10x (which means 10 times higher). Another thing is the use of “thrice”, which I have never seen although not wrong, indicating this manuscript was edited by someone with little scientific background. Please use “three times'.
- The returned pdf file has over 100 comments and corrections. Please address them carefully during revision to further strengthen the manuscript.
